

# obspyDMT: A Python Toolbox for Retrieving and Processing of Large Seismological Datasets

Kasra Hosseini[1,2] and Karin Sigloch[1]

[1]Dept. of Earth Sciences, University of Oxford, South Parks Road, Oxford OX1 3AN, UK.
[2]Dept. of Earth Sciences, Ludwig-Maximilians-Universität München, Theresienstrasse 41, 80333 München, Germany.

*Correspondence to:* Kasra Hosseini (kasra.hosseinizad@earth.ox.ac.uk)

**Abstract.** We present *obspyDMT*, a free, open source software toolbox for the query, retrieval, processing and management of seismological data sets, including very large, heterogeneous, and/or dynamically growing ones. obspyDMT simplifies and speeds up user-interaction with data centers, in more versatile ways than existing tools. The user is shielded from the complexities of interacting with different data centers and data exchange protocols, and is provided with powerful diagnostic and plotting tools to check the retrieved data and meta-data. While primarily a productivity tool for research seismologists and observatories, easy-to-use syntax and plotting functionality also make obspyDMT an effective teaching aid. Written in the Python programming language, it can be used as a stand-alone command line tool (requiring no knowledge of Python) or can be integrated as a module with other Python codes. It facilitates data archival, pre-processing, instrument correction, and quality control – routine but non-trivial tasks that can consume much user time. We describe obspyDMT's functionality, design and technical implementation, accompanied by an overview of its use cases. As an example of a typical problem encountered in seismogram preprocessing, we show how to check for inconsistencies in response files of two example stations. We also demonstrate the fully automated request, remote computation, and retrieval of synthetic seismograms from IRIS DMC's Syngine webservice.

## 1  Introduction

Seismology is a data-rich science, and since the advent of global digital networks in the 1990's, the growth of seismological waveform data holdings at international data centers has constantly accelerated. The data avalanche is a blessing, but also poses challenges to the scientist who needs to find and process these waveforms. Which data are available at the various international data centers? How can subsets of interest be selected, downloaded, organized, pre-processed, instrument-corrected and quality-controlled in a manageable amount of user time? Quality control and instrument corrections are non-trivial tasks, requiring tools that provide adequate diagnostics to verify data integrity. Almost every data-driven work flow in seismology begins with these considerations. As a project progresses, local data holdings often need to be updated, repaired, or extended, including the troubleshooting of earlier failed requests; adding waveforms made available since initial retrieval; adding (meta-)data from



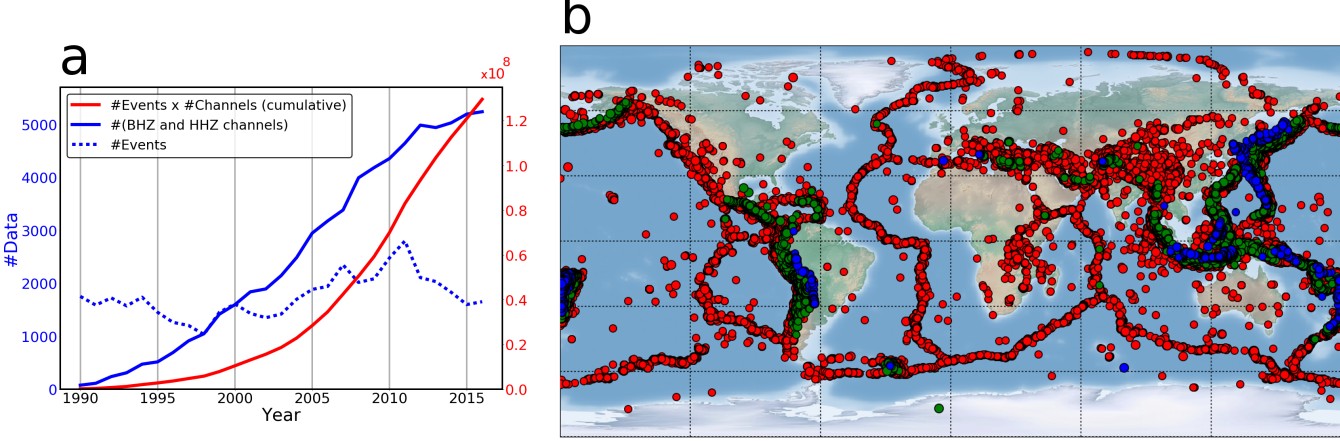

**Figure 1.** `obspyDMT --datapath iris_events_dir --min_date 1990-01-01 --max_date 2017-01-01 --min_mag 5.0 --event_info --plot_seismicity`

Rapid growth of seismological waveform data holdings at international data centers since 1990. Using the obspyDMT command above, we queried the IRIS DMC for hour-long, vertical, broadband (BHZ and HHZ) waveform segments containing earthquakes exceeding magnitude 5.0. (a) The data center's response. Red line shows cumulative sum of available event-based waveforms for this request, $\sum_{y=1990}^{year} [num\_events(y) \times num\_channels(y)]$. Number of events and seismograms in each year are shown by dotted and solid blue lines, respectively. (b) Global seismicity map of earthquakes in panel (a) colored by depth. Red: 0-70 km; green: 70-300 km; blue: ≥300 km. The generation of this map is triggered by the `--plot_seismicity` flag. Upon startup of the plotting module, the user can select the map style, "Shadedrelief" in this example.

other data centers; downloading corrected meta-data files. Surgical tasks of this kind can easily require more human supervision than the initial retrieval.

For a sense of data volumes, consider the example of Fig. 1, which arose in our work on global waveform tomography (Hosseini et al., 2014; Hosseini, 2016). Using the obspyDMT software, we queried the IRIS Data Management Center about hour-
5 long, broadband waveform segments containing earthquakes exceeding magnitude 5. Fig. 1a plots the data center's response: since 1990, IRIS' event catalog lists 1000-3000 such events per year, visualized in obspyDMT's automatically generated map of Fig. 1b. The number of archived broadband channels has grown to almost 5200 in 2016, and we are offered more than $10^8$ waveforms, corresponding more than 20 terabytes of data (and very long download times). Most applications would call for the selection of desirable subset of data before launching an actual request.
Besides large volumes, the hallmark of seismological data is heterogeneity. A culture of data sharing from permanent networks and temporary experiments means that waveforms get archived at many different data centers around the world; in different waveform and meta-data formats; documented and quality-controlled to varying degrees. Archives receive continuous inflows of data from telemetered stations, but also batchwise contributions from temporary experiments. Many experiments make meta-data available immediately but restrict access to actual waveforms for several years. No general mechanism exists
for broadcasting updates about data center holdings, which instead need to be actively and repeatedly queried by interested





users. Data access mechanisms tend to be specific to each center. Download of time-continuous or very long seismograms may be less supported than download of short segments around earthquake occurrences.

obspyDMT is free, open-source community software that strives to address these access challenges in a more comprehensive, integrated, and time-saving manner than existing software, which includes WILBER, WebDC, BREQ_FAST, NetDC, EMER-
ALD (West & Fouch, 2012), IGeoS (Morozov & Pavlis, 2011a, b), SOD (T. J. Owens & Oliver-Paul, 2004) and ObsPyLoad (Scheingraber et al., 2013). It is an easy-to-use command-line tool for the query, retrieval, and management of seismograms. The user is shielded from the complexities of interacting with different data centers, and provided with powerful diagnostic tools to check the retrieved data and meta-data, and to execute most routine pre-processing tasks, including instrument corrections. obspyDMT is written in the Python programming language, and runs on Linux, MacOS and Windows platforms.
Section 2 gives a high-level overview of obspyDMT's functionality in comparison to existing seismogram retrieval and management tools. Section 3 is a concise but near-complete tour that aims to turn the reader into a productive obspyDMT user very quickly, while also listing all usage options. Section 4 discusses implementation and performance of features that set obspyDMT apart from existing tools, specifically its communication with data centers, its robustness, and its diagnostics for instrument corrections.
All graphics in this paper were generated by obspyDMT. The caption of each figure gives the generating command(s) that handled the data and produced the plot.

## 2  Overview of software functionality

obspyDMT is a stand-alone tool for data retrieval and management that is not associated with any one seismological data center, data exchange protocol, or data format. In a style similar to Unix shells, it issues a single, one-line command
```
obspyDMT
```

which produces a default behavior and can be customized with many different options flags. There are no required options, and omission of an option flag will trigger a default behavior. This makes obspyDMT robust to run and friendly to learn. The
possibilities for customization are extensive, as will be discussed in Section 3. To give an idea, the command

```
obspyDMT --datapath iris_events_dir --min_date 1990-01-01 --max_date 2017-01-01 --min_mag
    5.0 --event_info --plot_seismicity
```

downloaded a global seismicity catalog from the IRIS DMC, saved the meta-data in a predefined directory structure, and generated Fig. 1 as a diagnostic display of the result. Invoking `obspyDMT` without any flags would have requested from the IRIS event catalog metadata for all events since 1970 that exceeded magnitude 3.0.

obspyDMT is part of the ObsPy ecosystem (Beyreuther et al., 2010; Megies et al., 2011; Krischer et al., 2015), an open-source community project that develops Python software for seismological observatories under the GNU Lesser General Public
License, hosted by the Ludwig-Maximilians-Universität Munich. obspyDMT uses many of ObsPy's utility functions, as well



as functions from Python's numpy, scipy and matplotlib libraries (Hunter, 2007), combining them into a more specialized piece of software. While no knowledge of Python is required to use obspyDMT, a software developer may seamlessly integrate it with other Python code. Python also makes it easy to wrap source codes written in other programming languages. For example, ObsPy wraps *evalresp*, IRIS' maturely developed software for instrument response corrections. obspyDMT's functionality can

be summarized as follows:

- Query of station meta-data: by absolute time or relative to earthquake occurrences; by geographic area (rectangles or circles); by channel or instrument type; wildcarding (*) is supported; simultaneous queries of different data centers.

- Query of earthquake source meta-data: from different catalog providers (currently from NEIC, Global CMT, IRIS DMC, NCEDC, USGS, INGV and ISC); event origin information or full moment tensors; by time window, region, event

magnitude and/or event depth.

- Diagnostic plots to visualize meta-data; plots are generated simply by appending an option flag to the data handling command.

- Retrieval of actual waveform data (seismograms) according to the results of meta-data queries. Support for different data exchange protocols (FDSN webservices, ArcLink).

- Retrieval of time-continuous series of arbitrary length; generation of diagnostic log files.

- Parallelized retrieval of waveform data from a data center for increased speed. Simultaneous retrieval from different data centers.

- Update mode: identical or modified queries can be relaunched; only new, modified, or previously failed data will be retrieved from the data center(s).

- Tolerant of retrieval errors and missing data (includes diagnostic logs).

- Automatic organization of data, meta-data and log files into standardized directory trees. (At present no tie to a database system.)

- Processing of retrieved data sets using default or user-defined instructions. ObsPy, SAC (George Helffrich & Bastow, 2013) or any other processing tool can be used to customize the processing unit on the waveform level. Supports pro-

25 cessing immediately upon waveform retrieval or later, batch-type processing. Support for parallel processing.

- Application of instrument responses. Support for various instrument formats (e.g., StationXML and Dataless SEED). Diagnostic plots of analog and digital filter stages. Option of parallelized instrument correction, taking advantage of multi-core architectures now common even on desktop processors.

- Automated retrieval of synthetic seismograms from IRIS' data services products (Hutko et al., 2017) for comparison to

30 real data.



**Table 1.** Comparison of seismological data retrieval and management tools. Abbreviations: E: event-based; C: continuous time series; U: update mode. obspyDMT is the only tool to provide access to both FDSN and ArcLink (in a single command); to retrieve both event-based and time-continuous waveform data; and to offer an "Update" mode for waveforms, response files and/or meta-data information. Few other tools provide for the management of data download and archival, instrument correction, or diagnostics plots.

| tool | method | data sources/interfaces | retrieval modes | archiving | instrument correction | plots |
|------|--------|-------------------------|-----------------|-----------|-----------------------|-------|
| WILBER | web portal | IRIS DMC or ODC/EIDA | E | ✗ | ✗ | ✗ |
| WebDC | web portal | ODC/EIDA | E | ✗ | ✗ | ✗ |
| BREQ_FAST | e-mail | IRIS DMC or ODC/EIDA | C | ✗ | ✗ | ✗ |
| NetDC | e-mail | NCEDC | C | ✗ | ✗ | ✗ |
| EMERALD | direct | IRIS DMC | E | ✓ | ✗ | ✓ |
| IGeoS | direct | IRIS DMC | E | ✗ | ✗ | ✓ |
| SOD | direct | FDSN | E | ✗ | ✓ (gain correction) | ✓ |
| obspyDMT | direct | FDSN and ArcLink | C, E, U | ✓ | ✓ | ✓ |

Header spanning: "data access" spans method, data sources/interfaces, retrieval modes; "data management" spans archiving, instrument correction, plots.

Various community software packages exist for achieving these tasks, but to our knowledge no other freely available package achieves them all. Table 1 compares the features of popular seismological community software to those of obspyDMT. We consider only tools that include functionality for data retrieval.

All data centers offer such tools, but each limited to retrieving data from that specific center. For example, both IRIS Data Management Center (DMC) in the U.S. and ORFEUS Data Center (ODC) in Europe implement the web-form-based "WILBER" service for retrieving event-based waveforms, as well as the e-mail-based "BREQ_FAST" service for time-continuous waveforms. If a user requires data from both centers, they need to be contacted separately. If event-based as well as continuous data are required, any given center needs to be contacted twice, using two different tools.

obspyDMT is the only tool among those in Table 1 that provides access to several data centers (in a single command), and to both types of waveform data (in two separate commands). The demand for continuous time series, often in large quantities, has surged with the rapid rise of cross-correlation methods based on ambient noise (Shapiro & Campillo, 2004). obspyDMT provides more convenient access than the e-mail-based tools BREQ_FAST or NetDC.

obspyDMT is also the only tool to offer an "update" mode for waveforms, response files, and/or meta-data information: re-launching a previous request will identify and retrieve only data that could not be retrieved earlier. Like obspyDMT, the SOD, IGeoS, and EMERALD tools are standalone software that runs on the user's computer rather than a data center server. All four communicate with data centers via the relatively new "Webservices" interfaces defined by the International Federation of Digital Seismograph Networks (FDSN). Queries are formulated as URL strings (Uniform Resource Locators) that point to physical data resources over the internet. We refer to this access method as "direct". Compared to older access methods, it can save much human intervention time by freeing the user of clicking through web pages (WILBER, WebDC) or managing





e-mails (BREQ_FAST, NetDC). SOD, IGeoS and EMERALD retrieve event-based waveforms only, i.e., queries are based on earthquake occurrences.

The standalone tools obspyDMT and EMERALD additionally manage the data download and archival to a local computer, thus relieving users of additional tedious and time-consuming steps. Both include certain plotting option, more extensively in

5 obspyDMT.

obspyDMT also offers full instrument correction based on RESP or StationXML station metadata, combined with diagnostic plots of transfer functions for individual filter stages. SOD is the only other tool to offer instrument correction, but this includes gain correction only, and it offers no diagnostic plots.

obspyDMT is the only tool to provide an automated update functionality for a user's existing, local data holdings.

## 10  3   Guided tour of use cases

The purpose of this section is to turn the reader into a proficient user of obspyDMT in the short space of a few pages. We demonstrate the most common use cases for the query, selection, retrieval and management of seismograms, metadata, and synthetic waveform. We list obspyDMT's full set of options in Table 2, which should be consulted as a cross-reference during the various stops of this guided tour.

We will:

1. Query event meta-data from different earthquake catalogs.

2. Query station meta-data from different data centers.

3. Request waveform data for a subset of events ("event-based mode"), from several different data centers.

4. Demonstrate how to update a local data set ("update mode").

5. Query and download continuous time series in arbitrary, user-provided time windows ("continuous mode").

6. Speeding up data retrieval by parallelization and bulk requests.

7. Demonstrate obspyDMT's plotting capabilities as we go.

8. Apply instrument corrections to waveform data.

9. Retrieve synthetic seismograms from Syngine webservice (Krischer et al., 2017), to match observed seismograms.

obpyDMT is a command-line tool that consists of a single command

```
obspyDMT
```

usually followed by option flags to modify the default behavior. Table 2 lists all available flag options, with explanations.

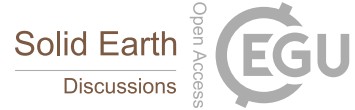

Table 2: Complete list of option flags to customize the default behavior of the `obspyDMT` command.

| Group | options | description | example |
|---|---|---|---|
| **check installation** | --help | Show this help message and exit. | |
| | --tour | Run a quick tour. | |
| | --check | Check all basic dependencies and their installed versions on the local machine and exit. | |
| | --version | Show the obspyDMT version and exit. | |
| **local path specification** | --datapath <PATH> | Path where obspyDMT will store/process/plot data (default: "./obspydmt-data"). | "/desired/path" |
| | --reset | If the datapath is found, delete it before running obspyDMT. | |
| **data retrieval modes** | --event_based | Event-based request mode (default). | |
| | --continuous | Continuous time series request mode. | |
| | --meta_data | Metadata request mode. | |
| | --local | Local mode for processing/plotting (no data retrieval). | |
| **general options (all modes)** | --data_source <SOURCE> | Data source(s) for retrieving waveform/response/metadata (default: 'IRIS'). | "IRIS" or "IRIS,ORFEUS" or "all" |
| | --print_data_sources | Print supported data centers that can be passed as arguments to --data_source. | |
| | --print_event_catalogs | Print supported earthquake catalogs that can be passed as arguments to --event_catalog. | |
| | --waveform <True/False> | Retrieve waveform(s) (default: True). | False |
| | --force_waveform | Retrieve waveform(s), force override of any pre-existing waveforms in local datapath directory. | |
| | --response <True/False> | Retrieve response file(s) (default: True). | False |
| | --force_response | Retrieve response file(s), force override of any pre-existing response files in local datapath directory. | |
| | --dir_select <DirNames> | Selects a subset of data directories for which to update/process/plot the contents (default False, i.e., all subdirectories will be considered). | "dir1,dir2" |
| | --min_epi <in deg> | Retrieve/plot all stations with epicentral distance $\geq$ min_epi. | "30" |
| | --max_epi <in deg> | Retrieve/plot all stations with epicentral distance $\leq$ max_epi. | "90" |
| | --min_azi <in deg> | Retrieve/plot all stations with azimuth $\geq$ min_azi. | "10" |
| | --max_azi <in deg> | Retrieve/plot all stations with azimuth $\leq$ max_azi. | "120" |
| | --list_stas <PATH> | User-provided station list instead of querying availability with a data center (default: False). | "/path/list-stations" |
| **time window, waveform format, and sampling rate (all modes)** | --min_date <DATE> | Start time, syntax: "YYYY-MM-DD-HH-MM-SS" or "YYYY-MM-DD" (default: '1970-01-01'). | "2010-09-24" |
| | --max_date <DATE> | End time, syntax: "YYYY-MM-DD-HH-MM-SS" or "YYYY-MM-DD" (default: Today). | "2015-01-01" |
| | --preset <in sec> | Time interval in seconds to add to the retrieved time series before its reference time. In event_based mode, the reference time is the earthquake origin time by default but can be modified by --cut_time_phase. In continuous mode, the reference time(s) are specified by --interval option, and --preset prepends the specified lead to each interval (default: 0). | "300" |
| | --offset <in sec> | Time interval in seconds to include to the retrieved time series after the time(s) reference. In event_based mode, the reference time is the earthquake origin time by default but can be modified by --cut_time_phase. In continuous mode, the reference time(s) are specified by --interval option, and --offset appends the specified offset to each interval (default: 1800). | "3600" |





Table 2: Complete list of option flags to customize the default behavior of the `obspyDMT` command.

| Group | options | description | example |
|---|---|---|---|
| | `--cut_time_phase` | In event_based mode, use as reference time the first-arriving phase (i.e., P, Pdiff or PKIKP, determined automatically). Overrides the use of origin time as default reference time. | |
| | `--waveform_format <mseed/sac>` | Format of retrieved waveforms. Default is miniseed ("mseed"), alternative option is "sac". This fills in some basic header information as well. | "sac" |
| | `--sampling_rate <in Hz>` | Desired sampling rate (in Hz). If not specified, the sampling rate of the waveforms will not be changed. | "10" |
| | `--resample_method <lanczos/decimate>` | Resampling method: "decimate" or "lanczos". Both methods use sharp low pass filters before resampling in order to avoid aliasing. If the desired sampling rate is 5 times lower than the original one, resampling will be done in several stages (default: "lanczos"). | "decimate" |
| **stations (all modes)** | `--net <NET>` | Network code (default: *). | "TA" or "TA,G" or "T*" or "*" |
| | `--sta <STA>` | Station code (default: *). | "RR01" or "RR01,RR02" or "R*" or "*" |
| | `--loc <LOC>` | Location code (default: *). | "00" or "*" |
| | `--cha <CHA>` | Channel code (default: *). | "BHZ" or "BHZ,BHE" or "BH*" or "*" |
| | `--identity <NET.STA.LOC.CHA>` | Identity code restriction, syntax: net.sta.loc.cha, e.g.: IU.*.*.BHZ to search for all BHZ channels in IU network (default: *.*.*.*). | "IU.*.*.BH*" |
| | `--station_rect <lonmin/lonmax/latmin/latmax>` | Include all stations within the defined rectangle, syntax: <lonmin>/<lonmax>/<latmin>/<latmax>. Cannot be combined with circular bounding box (`--station_circle`) (default: -180.0/+180.0/-90.0/+90.0). | "20/30/-15/35" |
| | `--station_circle <lon/lat/rmin/rmax>` | Include all stations within the defined circle, syntax: <lon>/<lat>/<rmin>/<rmax>. Cannot be combined with rectangular bounding box (`--station_rect`) (default: 0/0/0/180). | "20/30/10/80" |
| **speed up options (all modes)** | `--req_parallel` `--req_np <num_thread>` | Enable parallel waveform/response request. Retrieve several waveforms/metadata in parallel. Number of thread to be used in `--req_parallel` (default: 4). | "8" |
| | `--bulk` | Send a bulk request to a FDSN data center. Returns multiple seismogram channels in a single request. Can be combined with `--req_parallel`. | |
| | `--parallel_process` `--process_np <num_thread>` | Enable parallel local processing of the waveforms, useful on multicore hardware. Number of threads to be used in `--parallel_process` (default: 4). | "8" |
| **restricted data** | `--user <username>` `--pass <password>` | Username for restricted data requests, waveform/response modes (default: None). Password for restricted data requests, waveform/response modes (default: None). | "your_username" "your_password" |



Table 2: Complete list of option flags to customize the default behavior of the `obspyDMT` command.

| Group | options | description | example |
|---|---|---|---|
| **event-based mode** | `--event_catalog <CATALOG>` | Event catalog, currently supports LOCAL, NEIC_USGS, GCMT_COMBO, IRIS, NCEDC, USGS, INGV, ISC (default: LOCAL). | "IRIS" |
| | `--event_info` | Retrieve event information (meta-data) without downloading actual waveforms. | |
| | `--read_catalog <PATH>` | Read in an existing local event catalog and proceed. Currently supported catalog metadata formats: "QUAKEML", "NDK", "ZMAP". | "/path/to/file.ml" |
| | `--min_depth <in km>` | Minimum event depth (default: -10.0 (above the surface!)). | "10" |
| | `--max_depth <in km>` | Maximum event depth (default: +6000.0). | "100" |
| | `--min_mag <min_mag>` | Minimum magnitude (default: 3.0). | "4.0" |
| | `--max_mag <max_mag>` | Maximum magnitude (default: 10.0). | "7.0" |
| | `--mag_type <mag_type>` | Magnitude type. Common types include 'Ml' (local/Richter magnitude), 'Ms' (surface wave magnitude), 'mb' (body wave magnitude),'Mw' (moment magnitude), (default: None, i.e., consider all magnitude types in a given catalog). | "Mw" |
| | `--event_rect <lonmin/lonmax/latmin/latmax>` | Include all events within the defined rectangle, syntax: `<lonmin>/<lonmax>/<latmin>/<latmax>`. Cannot be combined with circular bounding box (`--event_circle`) (default: -180.0/+180.0/-90.0/+90.0). | "80/135/-15/35" |
| | `--event_circle <lon/lat/rmin/rmax>` | Search for all the events within the defined circle, syntax: `<lon>/<lat>/<rmin>/<rmax>`. Cannot be combined with rectangular bounding box (`--event_rect`) (default: 0/0/0/180). | "20/30/10/80" |
| **continuous time series mode** | `--interval <in sec>` | Specify time interval for subdividing long continuous time series (default: 86400 sec). | "3600" |
| **local processing** | `--pre_process <name_process_unit>` | Process retrieved data based on processing instructions in the selected processing unit (default: 'process_unit'). | 'process_unit_sac' |
| | `--force_process` | Forces to run the processing unit on the local/retrieved data, overwriting any previously processed data in local datapath directory. | |
| | `--instrument_correction` | Apply instrument correction in the process unit. | |
| | `--corr_unit <DIS/VEL/ACC>` | Correct the raw waveforms for displacement in $m$ (DIS), velocity in $m/s$ (VEL) or acceleration in $m/s^2$ (ACC) (default: DIS). | "VEL" |
| | `--pre_filt (f1,f2,f3,f4)` | Apply a bandpass filter to the seismograms before deconvolution, syntax: 'None' or '(f1,f2,f3,f4)' which are the four corner frequencies of a cosine taper, default: '(0.008, 0.012, 3.0, 4.0)'. | "(0.008, 0.012, 3.0, 4.0)" |
| | `--water_level <in dB>` | Water level in dB for instrument response deconvolution (default: 600.0). | "300" |
| **synthetic seismograms** | `--syngine` | Retrieve synthetic waveforms using IRIS/syngine webservice. | |
| | `--syngine_bg_model <MODEL>` | Syngine background model (default: 'iasp91_2s'). | 'iasp91_2s' or 'prem_a_2s' |
| | `--print_syngine_models` | Print supported syngine models that can be passed as arguments to `--syngine_bg_model`. | |





Table 2: Complete list of option flags to customize the default behavior of the `obspyDMT` command.

| Group | options | description | example |
|---|---|---|---|
| | `--syngine_geocentric_lat <True/False>` | Requesting synthetic seismograms based on geocentric latitudes of events/stations (default: True). | False |
| plotting | `--plot` | Activates plotting functionality. | |
| | `--plot_sta` | Plot all stations found in the specified directory (`--datapath`). | |
| | `--plot_availability` | Plot all availabilities (potential seismometers) found in the specified directory (`--datapath`). | |
| | `--plot_ev` | Plot all events found in the specified directory (`--datapath`). | |
| | `--plot_focal` | Plot beachballs instead of dots for event locations. | |
| | `--plot_ray` | Plot the ray coverage for all station-event pairs found in the specified directory (`--datapath`). | |
| | `--create_kml` | Create a KML file for event/station/ray. KML format is readable by Google-Earth. | |
| | `--create_event_vtk` | Create a VTK file for event(s). VTK format is readable by Paraview. | |
| | `--plot_seismicity` | Create a seismicity map and some basic statistics on the results. | |
| | `--depth_bins_seismicity <in km>` | Depth bins for plotting the seismicity histogram (default: 10 km). | "5" |
| | `--plot_waveform` | Plot waveforms arranged by epicentral distance. | |
| | `--plot_dir_name <raw/processed/...>` | Directory name that contains the waveforms for `--plot_waveform` option flag, e.g.: `--plot_waveform processed` (default: raw). | "raw" |
| | `--plot_save <PATH>` | Path where plots will be stored (default: '.', i.e., the current directory). | '.' |
| | `--plot_format <png/jpeg/pdf/...>` | Image format of plots (default: 'png'). | "png" |
| | `--plot_lon0 <lon0>` | Central meridian (x-axis origin) for projection (default: 180). | "160" |
| explore instrument responses (stationXML files) | `--plot_stationxml` | Plot the contents of stationXML file(s), i.e. transfer function of filter stages, specified by `--datapath`. | |
| | `--plotxml_date <DATE>` | Datetime to be used for plotting the transfer function, syntax: "YYYY-MM-DD-HH-MM-SS" or "YYYY-MM-DD". If not specified, the starting date of the last channel in the stationXML will be used. | "2010-01-01" |
| | `--plotxml_output <DIS/VEL/ACC>` | Type of transfer function to plot: DIS/VEL/ACC (default: VEL). | "DIS" |
| | `--plotxml_allstages` | Plot all filter stages specified in response file. | |
| | `--plotxml_paz` | Plot only Poles And Zeros (PAZ) of the response file, i.e. the analog stage. | |
| | `--plotxml_plotstage12` | Plot only stages 1 and 2 of full response file. | |
| | `--plotxml_start_stage <stage>` | First stage in response file to be considered for plotting the transfer function (default: 1). | "1" |
| | `--plotxml_end_stage <stage>` | Final stage in response file to be considered for plotting the transfer function, (default: last stage given in response file or the 100th stage, whichever number is smaller). | "3" |
| | `--plotxml_min_freq <in Hz>` | Minimum frequency in Hz to be used in transfer function plots (default: 0.01). | "0.001" |
| | `--plotxml_map_compare` | Plot all stations for which instrument responses have been compared (PAZ against full response). | |
| | `--plotxml_percentage <percent>` | Percentage of the phase transfer function's frequency range to be used for checking the difference between methods. "100" will compare transfer functions across their entire spectral range, i.e. from `min_freq` (set by `--plotxml_min_freq`) to Nyquist frequency; "80" compares from min_freq to 0.8 times Nyquist frequency (default: 80). | "100" |
| others | `--email <email address>` | Send an email to the specified address after completing the job (default: False). | "email_address" |
| | `--arc_avai_timeout <in sec>` | Timeout (in sec) for sending a data availability query via ArcLink (default: 40). | "60" |
| | `--arc_wave_timeout <in sec>` | Timeout (in sec) for sending a waveform data or metadata request via ArcLink (default: 2). | "60" |





### 3.1 Querying earthquake meta-data

First, we request event information from one of several supported seismicity catalogs, without downloading any waveforms
yet.

```
obspyDMT --datapath neic_event_dir --min_date 1990-01-01 --max_date 2017-01-01 --min_mag 5.0
    --event_catalog NEIC_USGS --event_info --plot_seismicity
```

This `obspyDMT` command with seven option flags queries the NEIC catalog (`--event_catalog NEIC_USGS`) for
all events exceeding magnitude 5.0 (`--min_mag`) that happened between 1990 and 2016 (`--min_date, --max_date`).
`--plot_seismicity` triggers the generation of the global seismicity map plot of Fig. 2. `--event_info` switches off
the retrieval of actual seismograms so that only meta-data are downloaded to a local directory named `neic_event_dir/`
(argument of `--data_path`). This directory is created if necessary, and is populated with the following subdirectory and
files:

```
neic_event_dir
└── EVENTS-INFO
    ├── catalog.txt
    ├── catalog.ml
    ├── catalog_table.txt
    ├── event_list_pickle
    └── logger_command.txt
```

Geographical restrictions for event (or station) queries are supported in rectangular or circular areas. For example, to extract
only earthquake meta-data for Indonesia, specify `lonmin/lonmax/latmin/latmax` as

```
--event_rect 80/135/-15/35
```

Appended to the earlier command, this generates the map inset of Fig. 2, top right. Note the rendering of colored beach balls
(deepest seismicity in the foreground). The global map of Fig. 2 also plots beach balls rather than simple black dots, but they
do not become apparent at this zoom level.

### 3.2 Query of station meta-data

Let's say we plan to investigate earthquakes exceeding magnitude 6.0 that occurred in this Indonesian rectangle at depths above
100 km. We want to know which seismometers in the Global Seismic Network were operational to record them from 1 Feb to
1 Dec 2014. We issue the query:



**Figure 2.** `obspyDMT --datapath neic_event_dir --min_date 1990-01-01 --max_date 2017-01-01 --min_mag 5.0 --event_catalog NEIC_USGS --event_info --plot_seismicity`

Global seismicity map of archived earthquakes in NEIC catalog of magnitude more than 5.0 that occurred between 1990 and 2016. One command queried the NEIC catalog, stored and organized the retrieved information and generated the seismicity map. (No actual waveform data were queried in this example). The results of some basic statistics (magnitude and depth histograms) are also generated and plotted automatically (top-left panel). Note the rendering of colored beach balls in the map inset (deepest seismicity in the foreground). The global map also contains beach balls rather than just simple black dots, but they do not become apparent at this zoom level.





```
obspyDMT --datapath event_based_dir --min_date 2014-02-01 --max_date 2014-12-01 --min_mag
    6.0 --max_depth 100 --event_rect 80/135/-15/35 --event_catalog NEIC_USGS --net _GSN
    --cha BHZ --meta_data
```

The NEIC event catalog returns 16 matching earthquakes, meta-data for which is stored in 16 separate subdirectories of a local directory called `event_based_dir`. Each of the 16 event subdirectories holds a subdirectory called `availability.txt` to which meta-data was written describing the GSN seismometers that were operational during the event. (Refer to Appendix A and Fig. A1 for a graphic depicting the full directory structure created by obspyDMT.) Only

10 station meta-data are requested, as specified by the mode flag `--meta_data`. We want StationXML files for (all) stations in the GSN network (`--net _GSN`), but only for the broadband, high-gain, vertical components of these stations, as specified by channel flag `--cha BHZ`. A subset of stations could be specified by the `--sta` flag, which supports wildcarding *, like many obspyDMT options. Since the option is absent here, it defaults to `--sta *`, i.e. all stations in the _GSN network. (See Table 2 for defaults for all options). The underscore in `--net _GSN` marks this as a virtual network, whereas the two regular

networks IU and II would be queried by `--net "IU,II"`.

### 3.3 Requesting and retrieving waveform data in event-based mode

Next, we retrieve the actual BHZ seismograms from the GSN network that were recorded during the 16 Indonesian earthquakes identified in Section 3.2. In our earlier obspyDMT command, only a few option flags need to be changed:

```
obspyDMT --datapath event_based_dir --min_date 2014-02-01 --max_date 2014-12-01 --min_mag
    6.0 --max_depth 100 --event_rect 80/135/-15/35 --event_catalog NEIC_USGS --net _GSN
    --cha BHZ --preset 300 --offset 3600 --instrument_correction --data_source IRIS
```

`--data_source` specifies explicitly that the IRIS DMC should be contacted, although this would also be the default if the

25 flag were omitted. If the user is unsure, it is best to specify `--data_source all`, which prompts obspyDMT to contact all 20 supported data centers listed in Table 3, and probably more in the future. (The list can be inspected by invoking obspyDMT `--print_data_sources`.)

`--preset 300` and `--offset 3600` specify the retrieval of waveform time windows of 300 s before to 3600 s after the reference time. Since we are downloading in event-based mode, i.e., centered around earthquake occurrences, the reference time

defaults to the event origin time. This could be changed to the time of P-wave arrival by invoking `--cut_time_phase` (see Table 2), in which case each seismogram would have a different absolute start time. obspyDMT knows that it is downloading in event-based mode because this is its default mode; adding the flag `--event_based` would have made this explicit. (`--meta_data` mode was introduced in Section 3.2; the alternative modes of `--continuous` and `--local` will be demonstrated shortly.)



Issuing this single-line command is the only requirement on user time, everything else is done automatically. Specifically, obspyDMT will:

1. Request event information from the NEIC event catalog `--event_catalog NEIC_USGS`.

2. In the `--datapath event_based_dir`, create a subdirectory `EVENTS-INFO/` containing a local catalog of meta-data for the 16 matching events. Also in `--datapath`, create 16 event subdirectories, each containing a subdirectory tree (info/, resp/, raw/, processed/) as in Appendix A, Fig. A1.

3. Retrieve station meta-data for all GSN stations for the 16 events in StationXML format from the IRIS data center, and save these to subdirectories `resp/`.

4. Retrieve BHZ waveforms of 3900 s duration from all matching GSN stations in miniseed format and save to subdirectories `raw/`.

5. Run default pre-processing operations on the waveforms, consisting of removing means and trends, tapering, filtering, and deconvolving the instrument response (all customizable). The processed seismograms are save to subdirectories `processed/`.

6. Save additional log files on query success to subdirectories `info/`.

Note how user time remains limited to issuing a single command no matter how many earthquakes, stations, or waveforms are being requested. Our tests required no human intervention even for very large requests that took weeks to download and encountered various time-outs or missing data issues at the data centers (c.f. Section 4.2).

### 3.4 Update of existing waveform data sets

In the course of working with a waveform data set, it often becomes necessary to update. This could mean requesting the same data again (because part of the earlier request failed for some reason), or expanding the number of earthquakes, stations, or seismograms. obspyDMT aims to be smart about these various cases, and not to retrieve duplicates unless the users explicitly wants it to. We demonstrate typical use cases. They have in common that the local `--datapath directory` must remain identical to that of any earlier request.

If an earlier query encountered problems (e.g., connection down, time-outs) or if the user has reason to expect that the data centers have added more seismograms since (e.g., the embargo period of a temporal network has ended), then it suffices to relaunch the exact same request (which was saved in log file `EVENTS-INFO/logger_command.txt`):

```
obspyDMT --datapath event_based_dir --min_date 2014-02-01 --max_date 2014-12-01 --min_mag
    6.0 --max_depth 100 --event_rect 80/135/-15/35 --event_catalog NEIC_USGS --net _GSN
    --cha BHZ --preset 300 --offset 3600 --instrument_correction --data_source IRIS
```



**Table 3.** List of international data centers that can be currently accessed via FDSN and ArcLink interfaces of obspyDMT. This list is growing as more and more data centers can be accessed directly (as opposed to ftp or email-based methods). `obspyDMT --print_data_sources` lists all available data centers, and `--print_event_catalogs` lists all available event catalogs.

| interface | data-source | URL |
|---|---|---|
| FDSN | BGR | http://eida.bgr.de |
| | EMSC | http://www.seismicportal.eu |
| | ETH | http://eida.ethz.ch |
| | GEONET | http://service.geonet.org.nz |
| | GFZ | http://geofon.gfz-potsdam.de |
| | INGV | http://webservices.rm.ingv.it |
| | IPGP | http://eida.ipgp.fr |
| | IRIS | http://service.iris.edu |
| | ISC | http://isc-mirror.iris.washington.edu |
| | KOERI | http://eida.koeri.boun.edu.tr |
| | LMU | http://erde.geophysik.uni-muenchen.de |
| | NCEDC | http://service.ncedc.org |
| | NIEP | http://eida-sc3.infp.ro |
| | NOA | http://eida.gein.noa.gr |
| | ODC | http://www.orfeus-eu.org |
| | ORFEUS | http://www.orfeus-eu.org |
| | RESIF | http://ws.resif.fr |
| | SCEDC | http://service.scedc.caltech.edu |
| | USGS | http://earthquake.usgs.gov |
| | USP | http://sismo.iag.usp.br |
| ArcLink | Many European data centers | |

obspyDMT compares the newly obtained event and station meda-data to its local versions and downloads only holdings that differ.

If the user wants to update only certain events, then `--min_date`, `--max_date`, `--min_mag`, `--max_mag` and/or `--event_rect` can be adjusted (see Table 2 for other options). Similarly, if the new date-time window is not contained within the old one, then additional events might fit the criteria and their waveforms would be added in new event directories.

If all 16 pre-existing event directories are to be updated, an alternative to the above command is to remove all event criteria, because obspyDMT will then default to the local, pre-existing event catalog in `EVENTS-INFO/` for earthquake meta-data.

```
obspyDMT --datapath event_based_dir --net _GSN --cha BHZ --preset 300 --offset 3600
    --instrument_correction --data_source IRIS
```





If the user decides he needs seismograms for all BHE channels (in addition to BHZ), the update command would be:

```
obspyDMT --datapath event_based_dir --net _GSN --cha BHE --preset 300 --offset 3600
    --instrument_correction --data_source IRIS
```

Augmenting the existing 16 events with seismograms from additional data centers is also an update operation because the waveform holdings of data centers often overlap to some extent. Again obspyDMT will automatically compare meta-data in order to avoid downloading duplicates. To update the data set with all vertical broadband channels of the GFZ and ORFEUS data centers, we would request:

```
obspyDMT --datapath event_based_dir --cha BHZ --preset 300 --offset 3600
    --instrument_correction --data_source "GFZ,ORFEUS"
```

`--datapath event_based_dir` is identical to what we defined in the previous command line that specifies the name of the top directory.

## 3.5    Retrieval of waveform data in time-continuous mode (`--continuous`)

In contrast to the examples thus far, some usage cases require waveforms that are not relative to or centered on specific earthquake occurrences. We refer to this usage mode as "time continuous" (`--continuous`). For example, studies that cross-correlate ambient noise often require long time series from many stations, often divided into segments of shorter duration (i.e., one day). ObspyDMT makes the handling of continuous time series easy, even if the data sets are voluminous.

```
obspyDMT --continuous --datapath yv_continuous_dir --min_date 2012-12-15 --max_date
    2013-01-15 --net YV --sta "RR0*,RR1*,RR2*" --cha BHZ --sampling_rate 10 --data_source
    RESIF --user your_username --pass your_password
```

This command queries the French RESIF data center for time series from 15 December 2012 to 15 January 2013 recorded by the temporary ocean-bottom seismometer network of the RHUM-RUM experiment (network code YV) (Barruol & Sigloch, 2013; Stähler et al., 2016). The wildcard "*" is used to specify multiple station names. Since the data are embargoed until the end of 2017, a username and password needed to be passed to the data center (`--user, --pass`). Here we were interested in noise levels on the ocean floor during the passage of tropical storm "Dumile" and therefore requested waveforms for the storm period, highlighted by the yellow box in Fig. 3. The storm was clearly recorded by elevated noise levels, whose variable onset times track the storm's diachronous passage across the $1500km \times 1500km$ wide network (Davy et al., 2014).

Long time series often need to be downsampled for ease of storage and handling, in this case to 10 Hz from originally 50 Hz (`--sampling_rate 10`). obspyDMT uses ObsPy functionality for resampling to any rate; if the frequency ratio is large, antialiasing and downsampling are automatically done in multiple stages.




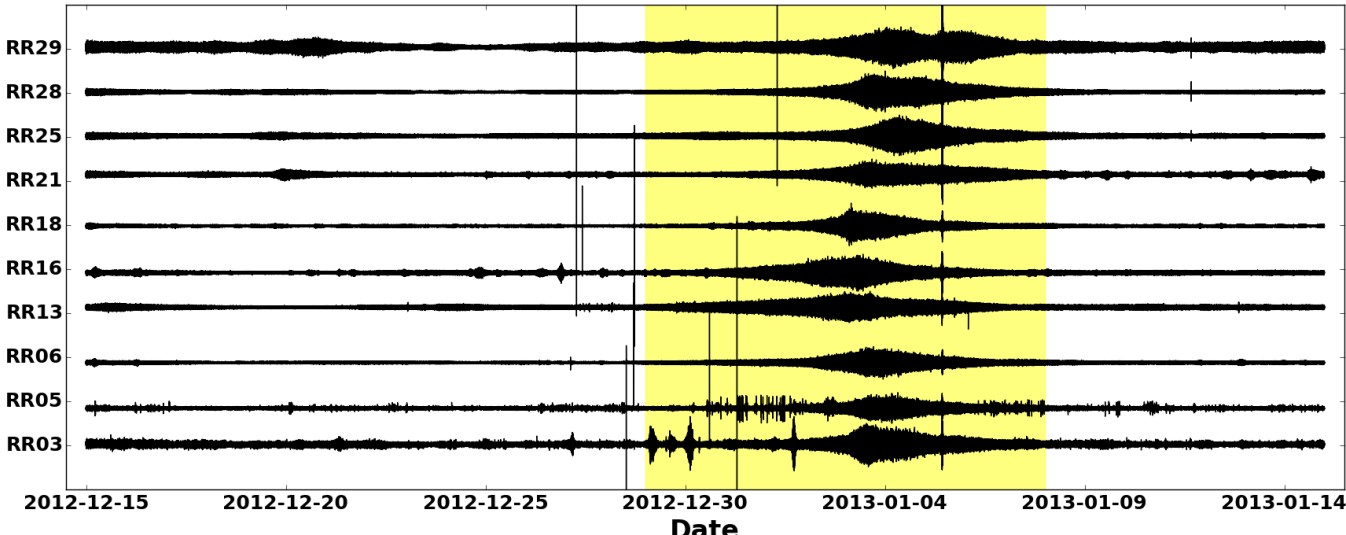

**Figure 3.** `obspyDMT --continuous --datapath yv_continuous_dir --min_date 2012-12-15 --max_date 2013-01-15 --net YV --sta "RR0*,RR1*,RR2*" --cha BHZ --sampling_rate 10 --data_source RESIF --user your_username --pass your_password`
Retrieval of continuous time series of arbitrary length, here for 30 days in 2012/2013. Data are from the temporary ocean-bottom network RHUM-RUM (network YV, station names RR*), which are currently still password-protected at the RESIF data center (`--user`, `--pass`). The command specifies downsampling to 10 Hz immediately upon retrieval. Passage of tropical storm "Dumile" is highlighted by the yellow box.

## 3.6 Speeding up data retrieval by parallelization

obspyDMT uses *ObsPy* clients to retrieve meta-data and actual waveforms from the data centers. Every request consists of three basic steps: 1) connect and send the data request to the data center; 2) download the data; 3) disconnect. By default, obspyDMT executes these steps for every meta-data or waveform request separately, e.g. $3 \times 1000$ steps if 1000 waveforms are requested. For large requests, this can become a serious bottleneck. To increase the efficiency in such cases, a functionality for parallelized data retrieval can be enabled as follows:

```
--req_parallel --req_np 4
```

The first flag changes the data retrieval mode from serial (default) to parallelized, and the second flag specifies the number of parallel requests.

The parallelization in obspyDMT is implemented on two levels: data center and waveforms. As an example of the former, if waveform data from both "ORFEUS" and "IRIS" are requested, obspyDMT sends parallel requests to these data centers.

The other parallelization is at waveform level: if several waveforms are requested from one data center, they are retrieved by `--req_np` parallel processes. (A good choice for np is the number of cores on the retrieving computer, i.e., 4 to 16 for





many current laptops or desktops.) The number of requested waveforms or meta-data files will be divided into the number of specified processes. Each process then sends and retrieves its set of request serially, but all processes organize their data into the same `--datapath` directory.

A further speed-up can be achieved by specifying a bulk request (`--bulk` flag). Instead of requesting individual items, this

will send a list of items (time series or meta-data) to the data center, which reduces the number of (dis-)connections. We have however noticed occasional instabilities (for very large requests, fewer waveforms retrieved than in serial mode), hence serial is set as the conservative default.

### 3.7   Plotting tools

obspyDMT offers various plotting tools for visualizing data sets. Fig. 2 demonstrated the plotting of seismic sources (beach

balls) on a map, via the `--plot_seismicity` option.

Fig. 4 demonstrates a map plot of ray paths between sources and receivers for the Indonesian example data set of Sections 3.1 to 3.4 in Google Earth:

```
obspyDMT --datapath event_based_dir --local --plot_ev --plot_focal --plot_sta --plot_ray
    --create_kml
```

Triggered by the plotting options, obspyDMT plots the contents of data directory "event_based_dir/", specifically the 16 event locations (`--plot_ev`) including focal mechanisms (`--plot_focal`); stations (`--plot_sta`); and ray paths

(`--plot_ray`). One file in KML format is created (`--create_kml`) which can be displayed by Google Earth. If `--create_kml` is omitted, obspyDMT plots the contents of the data set in maps similar to Fig. 2 or Fig. 5 (refer to Section 3.9). Flag `--local` explicitly tells obspyDMT to operate on pre-existing content in the local datapath directory, rather than making new contact with a data center.

### 3.8   Processing and instrument correction

obspyDMT can process the waveforms directly after retrieving the data, or it can process an existing data set in a separate step (local mode). By default, obspyDMT follows processing instructions described in the `process_unit.py` file located at `/path/to/my/obspyDMT/obspyDMT` directory. Although this file is fully customizable, several common processing steps can be done via options flags (without changing/writing new processing instructions). This includes:

1. resampling time series, for example, downsampling for ease of storage and handling (refer to Section 3.5 and

`--sampling_rate` option flag).

2. converting the format of retrieved waveforms to "SAC" and fill in some headers by simple inclusion of `--waveform_format sac` option flag.





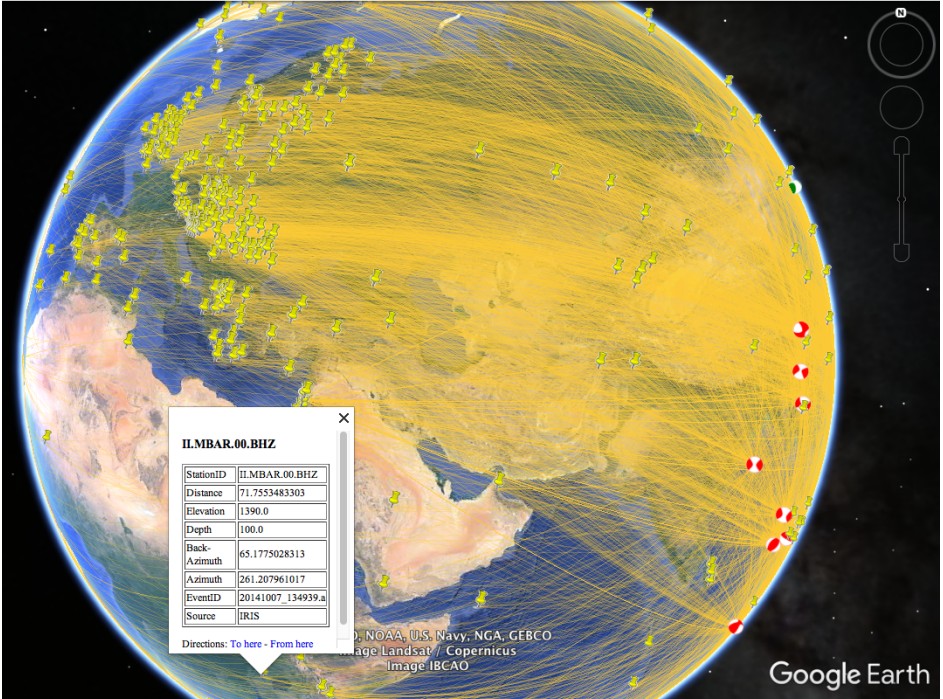

**Figure 4.** `obspyDMT --datapath event_based_dir --local --plot_ev --plot_focal --plot_sta --plot_ray --create_kml`
Plot of the contents of the `--datapath event_based_dir` that contains the Indonesian example data set generated in Sections 3.1 to 3.4. `--local` specifies that the existing, local waveform holdings should be plotted, rather than contacting the data centers anew. 16 earthquake locations are plotted as beachballs, stations featuring BHZ channels as yellow markers. Waveforms were retrieved from three data centers (IRIS, ORFEUS, GFZ).

3. instrument correction which includes removing means and trends, tapering, pre-filtering (customizable by `--pre_filt` option flag) and deconvolving the instrument response to displacement, velocity or acceleration (all customizable).

As an example, to correct the waveforms for instrument response directly after retrieving the data (similar to the example of Section 3.3):

```
obspyDMT --datapath event_based_dir --min_date 2014-02-01 --max_date 2014-12-01 --min_mag
    6.0 --max_depth 100 --event_rect 80/135/-15/35 --event_catalog NEIC_USGS --net _GSN
    --cha BHZ --preset 300 --offset 3600 --instrument_correction --data_source IRIS
    --corr_unit VEL
```



`--corr_unit VEL` specifies the physical unit of the processing output, in this case ground velocity in m/s. The same data set can be corrected for displacement in a separate step (not directly after retrieving the data):

```
obspyDMT --datapath event_based_dir --local --force_process --instrument_correction
       --corr_unit DIS
```

Since obspyDMT stores processed waveforms in the `processed` directory (Fig. A1), a good practice is to re-name all `processed` directories before launching the above command line; otherwise, previously processed waveforms will be over-
written (`--force_process`).

The user can also modify the `process_unit.py` or write a new script with new processing instructions. Currently, these files need to be located in the `/path/to/my/obspyDMT/obspyDMT` directory and can be accessed via `--pre_process my_proc_unit` option flag, replacing `my_proc_unit` with the name of the python script. The instructions are written at the waveform level, and obspyDMT automatically applies them to all archived waveforms. The main advantage of this design
choice is its flexibility. The user can customize the processing instructions using available tools in ObsPy; moreover, other processing tools can be used or combined to write these instructions. As an example, the following command line calls a processing instruction `process_unit_sac.py`, this file is located in `/path/to/my/obspyDMT/obspyDMT`:

```
obspyDMT --datapath event_based_dir --local --force_process --pre_process process_unit_sac
```

Here, SAC (instead of obspy) is used to remove the mean, apply a Hanning window, compute the FFT, plot the amplitude spectrum of each waveform on a log-log plot and save the images as pdf files in the `processed` directory.

## 3.9 Requesting synthetic seismograms

obspyDMT facilitates the generation of synthetic waveforms matching the real data in two ways by: 1) retrieving synthetic
waveforms from a new IRIS webservice: Syngine (Krischer et al., 2017) 2) providing required meta-data for calculating synthetic waveforms using external tools.

Syngine delivers fully numerical seismic waveforms computed on common spherically symmetric Earth models (PREM, ak135-f, IASP91). The following example command retrieves not only observed waveforms but also their synthetic counterparts, computed on a PREM (Dziewonski & Anderson, 1981) anisotropic background model:

```
obspyDMT --datapath data_fiji_island --min_mag 6.8 --min_date 2014-07-21 --max_date
       2014-07-22 --event_catalog NEIC_USGS --data_source IRIS --min_azi 50 --max_azi 55
       --min_epi 94 --max_epi 100 --cha BHZ --instrument_correction --syngine
--syngine_bg_model prem_a_2s
```



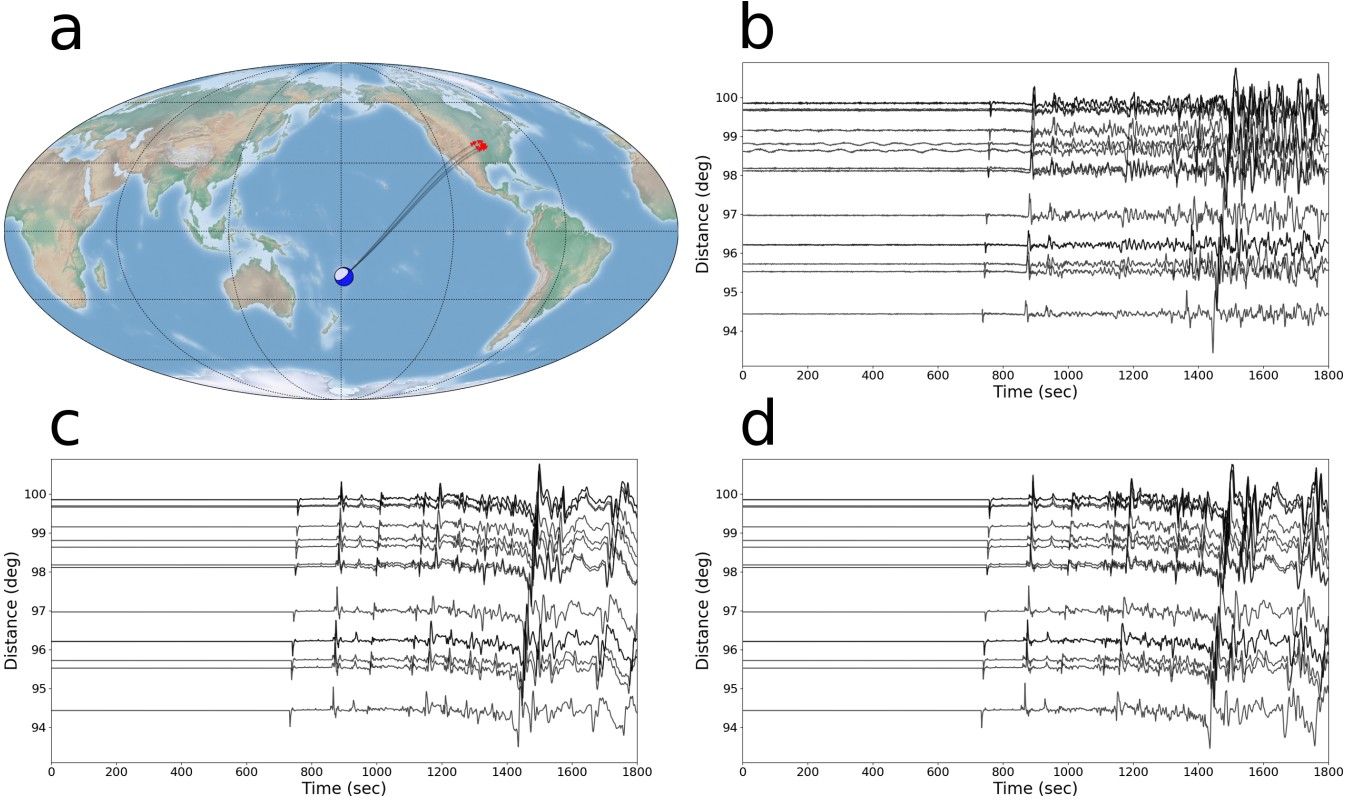

**Figure 5.** Observed versus modeled broad-band seismograms for an earthquake of magnitude 6.9 *Mw* in the Fiji Islands Region
(2014/07/21 14:54:41, at 19.802°S, 178.4°W, 615 km depth). (a) Source and receiver distribution plotted by `obspyDMT --datapath`
`data_fiji_island --local --plot_ev --plot_focal --plot_sta --plot_ray`. Note the distribution of stations
with respect to the event. The options flags `--min_azi`, `--max_azi`, `min_epi` and `max_epi` specified minimum azimuth, maximum az-
imuth, minimum distance and maximum distance for station search, respectively. (b) Observed broad-band waveforms plotted by `obspyDMT`
`--datapath data_fiji_island --local --plot_waveform --plot_dir processed`. (c) Synthetic seismograms re-
trieved from the Syngine webservice for a PREM anisotropic background model. The stored waveforms are plotted by `obspyDMT`
`--datapath data_fiji_island --local --plot_waveform --plot_dir syngine_prem_a_2s`. (d) similar to (c)
except for IASP91 background model. Plotted by `obspyDMT --datapath data_fiji_island --local --plot_waveform`
`--plot_dir syngine_iasp91_2s`.



Two option flags that triggered the synthetic waveform retrieval are: `--syngine` and `--syngine_bg_model prem_a_2s`. The option flags `--min_azi`, `--max_azi`, `min_epi` and `max_epi` specify minimum azimuth, maximum azimuth, minimum distance and maximum distance for station search, respectively. The synthetic waveforms are stored in `syngine_prem_a_2s` directory, the contents of which can be plotted by obspyDMT plotting tools (refer to Fig. 5).

5    Changing the argument of `--syngine_bg_model` to `iasp91_2s`, synthetic seismograms based on `IASP91` (Kennett & Engdahl, 1991) background model can be retrieved (Fig. 5):

```
obspyDMT --datapath data_fiji_island --min_mag 6.8 --min_date 2014-07-21 --max_date
    2014-07-22 --event_catalog NEIC_USGS --data_source IRIS --min_azi 50 --max_azi 55
    --min_epi 94 --max_epi 100 --cha BHZ --instrument_correction --syngine
    --syngine_bg_model iasp91_2s
```

All earth reference models currently supported by Syngine can be listed by invoking:

```
obspyDMT --print_syngine_models
```

Alternatively, meta-data information and logfiles generated and organized by obspyDMT can be used to link an archived data set to other software for the generation of synthetic seismograms. A practical example for this is multiple-frequency tomography. In this method, frequency-dependent observables (phase shifts or amplitudes) are measured by cross-correlating the recorded waveforms with the corresponding synthetic seismograms in multiple frequency bands (Sigloch, 2008; Zaroli et al., 2015; Hosseini & Sigloch, 2015). Synthetic seismograms need to be computed for exactly the same sources and receivers in the data set. This includes source characteristics (epicenter, depth, moment tensor and source time function) and receiver specifications (latitude, longitude, elevation and burial).

obspyDMT stores station information in one ASCII file per event and in the SAC headers (if this waveform format is selected). It automatically updates meta-data information and logfiles of a local data archive if stations are added/removed. Event information is written in QuakeML and ASCII formats. Although basic source and receiver information can be retrieved from most data centers, moment tensor solutions are available only in certain seismicity catalogs, among them the NEIC and GCMT catalogs, which are both supported by obspyDMT (refer to moment tensor retrieval as demonstrated by Fig. 2).

In summary, obspyDMT retrieves, organizes and stores all meta information required to compute synthetic seismograms using arbitrary forward modeling tools. Users only need to provide scripts that connect this meta-data input to their desired computational engine (other than Syngine), for example, *AxiSEM* (Nissen-Meyer et al., 2014) or *Instaseis* (van Driel et al., 2015).





## 4    Discussion

Here we discuss implementation and performance issues, specifically obspyDMT's communication with data centers; its robustness in case of large and heterogeneous requests; and the usefulness of the instrument correction diagnostics. All three features set obspyDMT apart from existing tools.

### 4.1    Communication with data centers

obspyDMT can retrieve data from a multitude of international data centers (c.f. Table 3, a list that is growing). The user is shielded from having to know communication specifics for each data center. Under the hood, the software implements ObsPy clients for two different kinds of data exchange protocols: FDSN Web Services and ArcLink.

In 2013, the International Federation of Digital Seismograph Networks (FDSN) defined common Web Services interfaces (http://www.fdsn.org/webservices/), allowing data request tools to work with any of the growing number of FDSN data centers that implement these interfaces (http://www.fdsn.org/webservices/datacenters/). These centers currently include the IRIS DMC, BGR, EMSC, ETH, GEONET, GFZ, INGV, IPGP, ISC, KOERI, LMU, NCEDC, NIEP, NOA, ODC, ORFEUS, RESIF, SCEDC, USGS, and USP. Three service interfaces are specified by the FDSN and supported by obspy: fdsnws-station for accessing station meta-data in StationXML format; fdsnws-dataselect for accessing time series in miniseed format; and fdsnws-event for accessing earthquake parameters in in QuakeML format. obspyDMT offers conversion to other formats, e.g., SAC for waveforms `--waveform_format sac`. Requests are sent via the HTTP internet protocol for individual requests and via HTTP-POST for lists of requests, so that data can be requested from any web browser by generating URLs.

ArcLink is an older data request protocol that arose in Europe in order to virtually consolidate distributed seismological data holdings across various European countries. It is a distributed request protocol developed by the German WebDC initiative of GEOFON and BGR (Bundesanstalt für Geowissenschaften und Rohstoffe) as a continuation of the NetDC concept originally developed by the IRIS DMC. ArcLink communicates via TCP/IP rather than via supervision-intensive e-mail or ftp requests required by other access mechanisms at the time. It accesses waveform data in MiniSEED or SEED format, and associated meta information as Dataless SEED files. At the time we developed obspyLoad, a pre-cursor of obspyDMT (Scheingraber et al., 2013), only a few data centers were implementing FDSN Web Services. Hence ArcLink clients greatly expanded the reach of obspyLoad, to include most European data centers. obspyLoad contacts the ORFEUS DMC via ArcLink, which in turn "forwards" ArcLink requests to other data centers across Europe. This ArcLink functionality remains in obspyDMT, but if a data center implements both interfaces, then obspyDMT accesses it via Web Services (default), which now includes the European data centers. It seems likely that Web Services will completely supersede ArcLink.

### 4.2    Robustness of data retrieval

In our research we have used obspyDMT extensively, in order to retrieve several voluminous, event-based data sets for global-scale tomography, from different combinations of data centers. We have also requested large volumes of time-continuous data ("ambient noise") for cross-correlation studies. In all cases, we observed obspyDMT to work stably, i.e., requiring no





user intervention despite the fact that many individual waveform requests encounter errors from the data centers, for various reasons. obspyDMT caught all exceptions and continued undeterred.

In a demanding test that expanded the scope of the example of Section 3.3, we retrieved all BHZ channels from all supported data sources, in event-based mode, requesting earthquakes exceeding magnitude 6.0 that occurred during two years. The idea
was to test the most challenging request mode, which includes station and event meta-data, and to communicate with all data centers, including some that implemented Web Services very recently.

```
obspyDMT --datapath 2014_2015_dataset --min_date 2014-01-01 --max_date 2016-01-01 --min_mag
    6.0 --event_catalog NEIC_USGS --cha BHZ --data_source all --preset 300 --offset 3600
--req_parallel --req_np 8 --pre_process False
```

The retrieval took 2 days and 10 hours on a standard desktop with 4 CPUs. The retrieved data set was 145 GB in size, containing 293 events and 685,388 waveforms. No user intervention was required at any stage.

This finding is consistent with the performance of obspyDMT's predecessor obspyLoad (Scheingraber et al., 2013), obspy-
DMT's predecessor. With an event-based request similar to the one above, to all data centers available at the time (in 2012 this was IRIS and the European centers via ORFEUS/ArcLink), we retrieved 162 GB of waveform data, consisting of 690,503 MiniSEED files for three components (BHZ, BHE and BHN) for 154 events. The retrieval took 45 days because the job slowed considerably after the first 73 GB (but continued at the old speed after relaunching, i.e., requesting the remaining 89 GB through update mode). The fraction of successfully retrieved waveforms varied strongly between data centers and ranged from
99.8% to 34.8% (availabilities were verified by spot checks in manual retrieval attempts). The exact reasons for the slowdown remained unclear, but aside from the decision to relaunch, no user intervention was required at either download stage.

For the current test in 2017, no such slowdown was observed, and the retrieval of a comparable data volume (145 GB) took only a twentieth of the time (2.5 days), despite being routed to many more data centers. We conclude that obspyDMT works robustly with all supported data centers, even for large and heterogeneous data and meta-data requests.

**4.3 Instrument correction**

If station meta-data could be routinely trusted, correcting for instrument responses would amount to a simple series of deconvolutions, of a number of impulse responses (analogue and digital "filter stages" from raw waveforms). Unfortunately it is not uncommon for filter information in station meta-data files to be erroneous. Some of the resulting artifacts in the displacement or velocity seismograms are large enough to potentially cause serious geoscientific misinterpretation, such as pronounced
traveltime delays under an isolated island station where in reality there are none.

Problems with the contents of StationXML or SEED/RESP files may or may not be straightforward to identify, as discussed below. A full visual representation of filter impulse responses can greatly facilitate the trouble shooting. obspyDMT implements several plotting options for this purpose, as demonstrated in Section 3.8 and Figs. 6-8.

An instrument response typically consists of a first, analogue stage (a.k.a. "poles-and-zeros", or PAZ stage), which describes
the transfer function of the sensor, and several digital stages, which describe the A/D conversion, anti-aliasing, and downsam-





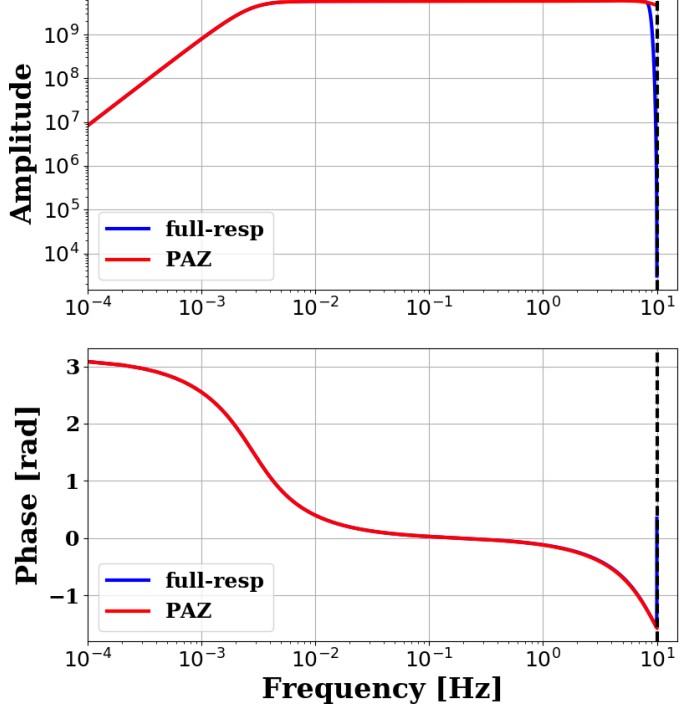

**Figure 6.** `obspyDMT --datapath /path/to/STXML.IC.XAN.00.BHZ --plot_stationxml --plotxml_paz`
`--plotxml_min_freq 0.0001`

Transfer function spectra (amplitude and phase) of a Streckeisen STS-1VBB w/E300 station (IC.XAN) in China. Blue lines show transfer function components computed for all filter stages in *StationXML* file; red lines are for the analogue part. The two functions match very well in all frequencies except for amplitude spectra close to the Nyquist frequency (dashed line).

pling inside the data logger. The PAZ stage is rarely problematic, whereas specifications of the digital stages are error-prone. Our discussion of neuralgic points and their possible diagnosis follows the PhD thesis of Groos (2010).

Coefficients of asymmetric FIR filters are sometimes given in reverse order from that expected by the SEED convention, which can cause erroneous time delays of up to 1 s in the "corrected" waveforms. This issue may not be easy to detect as it
5  requires knowledge of the correct order of filter coefficients, e.g., by comparing to a trusted StationXML file describing the same data logger in a different location.

A typical, unproblematic response resembles Fig. 6, with PAZ and full resp coinciding everywhere except near the Nyquist frequency. By contrast, a plot like Fig. 7 can flag up a potential problem. The very different phase responses of PAZ-only versus full response indicate that the digital stages introduce a significant delay (and possibly distortion) of the corrected time series.
10  The user can then question whether this behavior is expected from the data logger. obspyDMT automatically creates diagnostic reports for stations where PAZ and full response differ significantly. Fig. 8 further zooms in on the issue, by indicating that among the digital stages, only Stage 5 has a non-zero phases response, identifying it as the questionable one. If the user





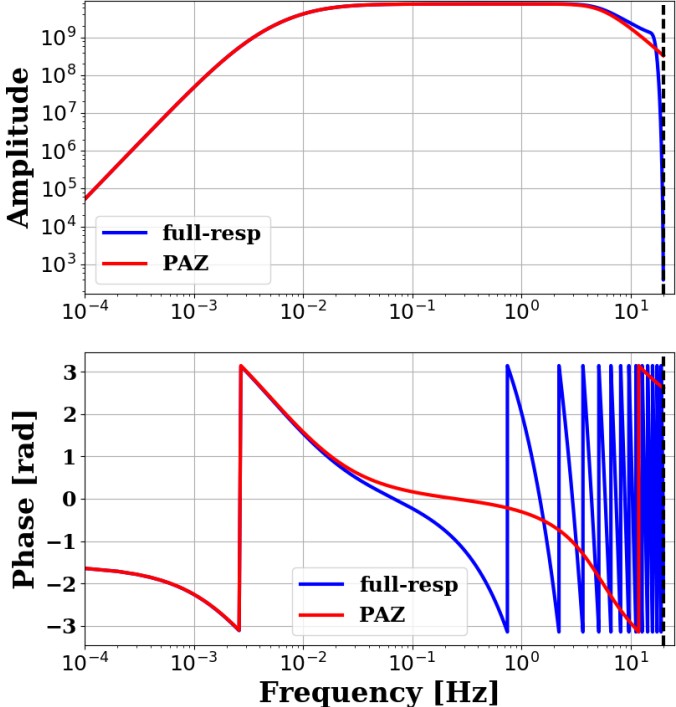

**Figure 7.** `obspyDMT --datapath /path/to/STXML.GT.LBTB.00.BHZ --plot_stationxml --plotxml_paz --plotxml_min_freq 0.0001`

Transfer function spectra (amplitude and phase) of a Geotech KS-54000 Borehole seismometer (GT.LBTB) in Botswana. Blue lines show transfer function components computed for all filter stages in the *StationXML* file; red lines are for the analogue part. A large discrepancy exists between the phase spectra of the two transfer functions. The deviation emerges at frequencies around $10^{-2}$ Hz and increases up to the Nyquist frequency. Fig. 8 shows that this difference is caused by one of the digital stages in the instrument response.

decides that the digital stage specifications are suspect, he can choose to apply PAZ-only correction rather than full response - this should give a decent result, except for frequencies very close to Nyquist. Alternatively, if the user is working with low-frequency data only (below 0.01 Hz), he can conclude that no problem would ever arise because even Stage 5 is almost zero in that spectral range.

5    Another recurring problem concerns delay time values specified for the FIR filter stages. According to the SEED manual, corrected filter delay times have to be positive; and yet, negative or zero values are sometimes encountered in retrieved meta-data files which can result in erroneous time shifts of 1 to 2 s in corrected waveforms. This problem is easily spotted, but seven years after the report by (Groos, 2010), we still encounter such response files delivered by data centers.

obspyDMT also checks for inconsistencies in the "Estimated delay" and the "Correction applied" of the digital filter stages.

10   In modern data loggers, these two values are usually similar, because delay times are removed from the waveforms internally. However, discrepancies have been observed, such as negative or zero values for the corrected delay time. In the example of





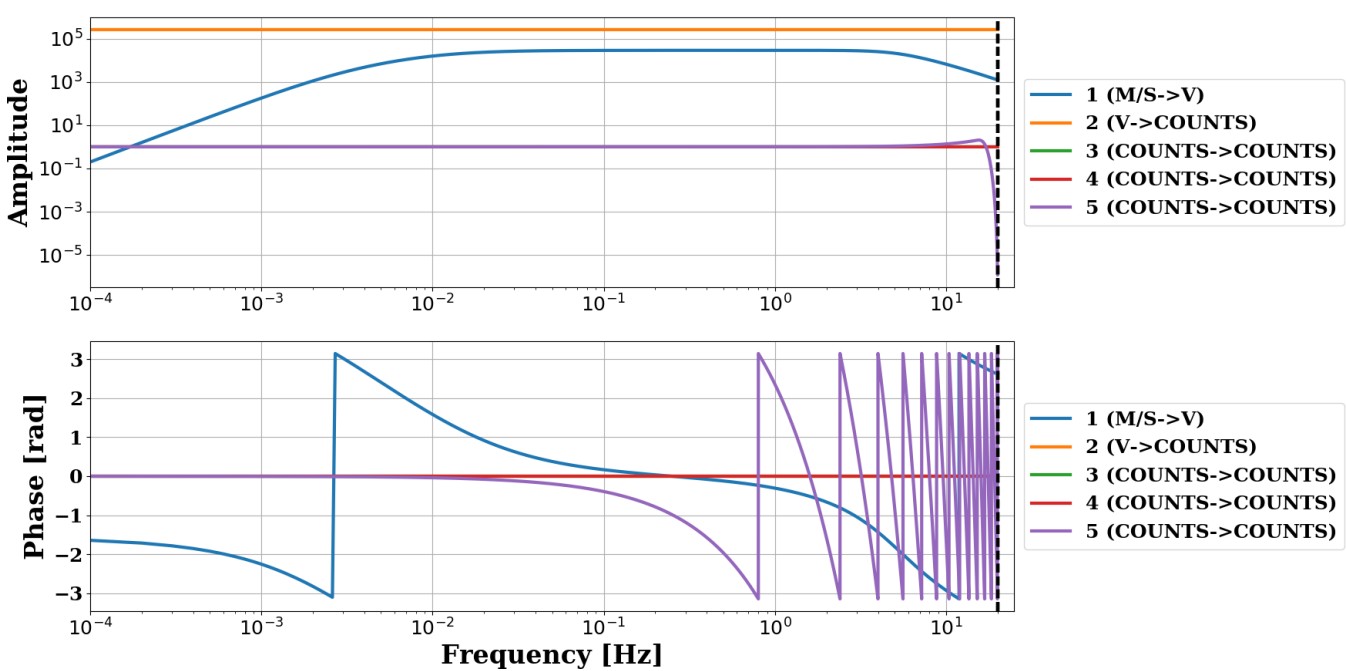

**Figure 8.** `obspyDMT --datapath /path/to/STXML.GT.LBTB.00.BHZ --plot_stationxml --plotxml_min_freq 0.0001 --plotxml_allstages` Transfer function spectra (amplitude and phase) of each stage in the *StationXML* file of a Geotech KS-54000 Borehole seismometer (GT.LBTB) in Botswana. In the phase response, two stages (1 and 5) have non-zero values. Both stages contribute to the phase spectrum of the complete instrument response ("full-resp") of Fig. 7. However, effects of stage 5 on amplitude and phase spectra are not considered in "PAZ" (analogue).

Fig. 7, the estimated delay is reported as 0.63 s, and the applied correction is 0.0 s. obspyDMT collects this information and automatically generates one diagnostic report for the results of all consistency checks.

## 5 Conclusions

We presented obspyDMT, a new software for query, retrieval, processing and management of large seismological data sets. Its

5   functionality, design and technical implementation were described and compared with existing seismological data retrieval and management tools. Through examples we demonstrated its main functionalities, such as query of station and earthquake source meta-data (full moment tensor and event origin), retrieval of event-based or time-continuous waveform data from various data centers in one command line, update mode, customizable processing unit, and automatic organization of (meta-)data and logfiles into standardized directory trees. The user is provided with powerful diagnostic and plotting tools to check the

10   retrieved data and meta-data. For large seismological data sets, data retrieval and processing can be parallelized on multi-core





architectures by simple inclusion of an option flag. Using obspyDMT's diagnostic plots of analogue and digital filter stages, we checked the spectra (amplitude and phase) of instrument response files. Synthetic seismograms matching an example data set were retrieved from IRIS syngine.

In all these use cases, issuing a single line command is the only requirement for the user, everything else is done automati-
5  cally.

Refer to Appendix C for instruction how to download and install obspyDMT.

*Acknowledgements.* We are grateful to Joachim Wassermann for detailed discussions on instrument correction. We thank Piero Poli for valuable ideas and discussions on obspyDMT functionalities. All waveform data used for the examples came from the IRIS, ORFEUS, GFZ and RESIF data management centers. K.H. was funded by Deutsche Forschungsgemeinschaft (DFG) grants made to K.S., grant numbers
10  SI 1538/1-1 (in Priority Programme SAMPLE) and SI 1538/2-1 (project RHUM-RUM). The research leading to these results has received funding from the People Programme (Marie Curie Actions) of the European Union's Seventh Framework Programme FP7/2007-2013/ under REA grant agreement n° PCIG14-GA-2013-631104 RHUM-RUM.



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





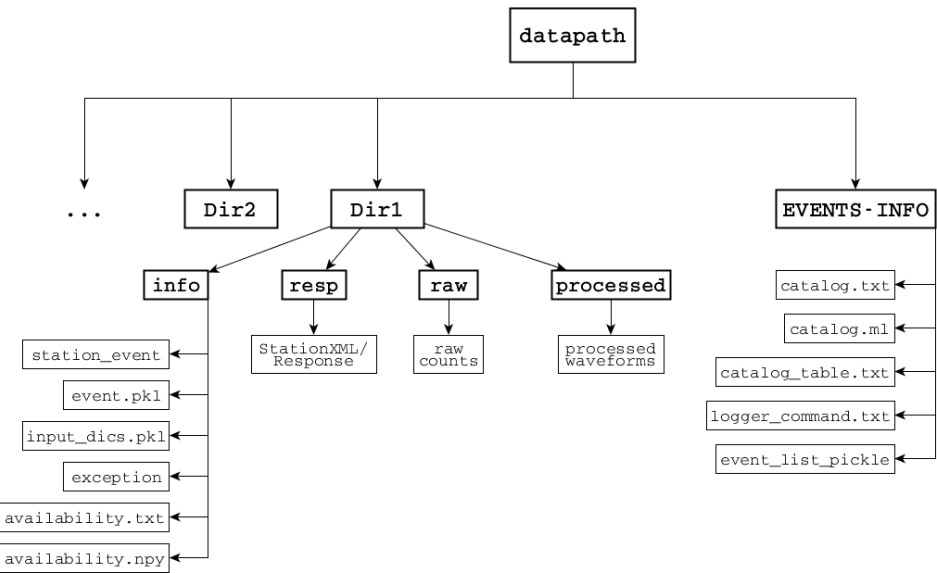

**Figure A1.** For each request, obspyDMT creates the depicted directory tree inside the user-specified directory `datapath/`, and arranges the retrieved data either in different event directories (for event-based requests) or in chronologically named directories (for continuous requests). It also creates a directory in which a catalog of all requested events/time spans is stored. Raw waveforms, StationXML/response files and corrected waveforms are collected in sub-directories. While retrieving the data, obspyDMT creates metadata files such as station/event location files, stored in the `info/` directory of each event.

## Appendix A: Directory structure

ObspyDMT organizes retrieved seismograms and metadata in a standardized directory structure, as shown in Fig. A1.

## Appendix B: Instrument correction

5  Seismograms recorded by digital broadband seismometers are stored as digitized voltage signals called *raw counts*. The relation between this signal and ground motion (e.g., displacement) depends on response functions of the seismometer and data logger components (sensor, amplifiers, A/D converters, digital filters). Each component is referred to as a "stage" characterized by a transfer function, and the entire system can be described by the cumulative transfer function, i.e. product in the frequency domain of the individual stage transfer functions. The instrument sensor, i.e. the analogue measurement apparatus before A/D




conversion, is referred to as the analogue stage or the "poles-and-zeros" stage. Following the nomenclature of *SEED Manual* (Ahern et al., 2012), its frequency response G can be written as

$$G(j\omega) = S_d A_0 \frac{\prod_{n=1}^{N}(j\omega - r_n)}{\prod_{m=1}^{M}(j\omega - p_m)} \tag{B1}$$

$r$ and $p$ stand for zeros and poles of a system. $N$ and $M$ are the number of zeros and poles, respectively. $S_d$ is the stage gain. $A_0$ is the normalization factor which scales the amplitude of the poles-and-zeros polynomial to unity at a reference frequency (usually 1 Hz):

$$A_0 \left| \frac{\prod_{n=1}^{N}(j\omega_{ref} - r_n)}{\prod_{m=1}^{M}(j\omega_{ref} - p_m)} \right| = 1 \tag{B2}$$

$G$ relates the ground motion $V$ (input signal) to recorded raw counts $R$ by: (Scherbaum, 1996)

$$R(j\omega) = G(j\omega) \times V(j\omega) \tag{B3}$$

in which, $R(j\omega)$ and $V(j\omega)$ are the Fourier transforms of raw counts and ground motion, respectively. Instrument response correction can be carried out by transforming the raw seismogram $R(t)$ to the spectral domain, dividing $R(j\omega)$ by $G(j\omega)$ (deconvolution in time) and transforming the result back into the time domain, in order to obtain $V(t)$ in the physical units of displacement, velocity, or acceleration.

Instrument responses are provided by data centers in different formats. An older format called SEED describes transfer functions of all analogue and digital stages in a seismometer and is hence sufficient to calculate the frequency response function of a seismic channel ($T(j\omega)$ in eq. B3). In practice, this format is usually converted to human readable ASCII files called *SEED RESP* that can be read by other instrument correction software such as *evalresp*. Recently, *FDSN* defined a new format *FDSN StationXML* which contains the most important and commonly used structures of *SEED* metadata in *XML* representation (FDSN, 2015). Compared to SEED, StationXML simplifies and adds clarification to station metadata. All data centers that support *FDSN* web services deliver instrument responses in this format. obspyDMT can read and interpret both StationXML and SEED.

### Appendix C: Installation and system requirements

### C1 ObsPy

ObsPy (Beyreuther et al., 2010; Megies et al., 2011; Krischer et al., 2015) is currently running and tested on Linux (32 and 64 bit), Windows (32 and 64 bit) and Mac OS X. Please refer to the ObsPy webpage for complete notes regarding ObsPy installation on different platforms.





In addition to Python and ObsPy tools, obspyDMT builds on `NumPy` an extension for performing numerical calculations on large arrays and matrices (van der Walt et al., 2011); `matplotlib` a popular plotting package (Hunter, 2007); `matplotlib basemap toolkit` (Whitaker, 2015) to project the data on a map and `SciPy` (Jones et al., 2001) a library for advanced math, signal processing or statistics. Most of these libraries are prerequisites for installing ObsPy and are used in obspyDMT.

## C2  obspyDMT

Once working Python and ObsPy environments are available, obspyDMT can be installed in different ways:

**1. install obspyDMT package locally (using PyPi):** which tends to be the most user-friendly option:

```
pip install obspyDMT
```

**2. install obspyDMT from the source code:** The latest version of obspyDMT is available on GitHub. After installing `git`:

```
git clone https://github.com/kasra-hosseini/obspyDMT.git /path/to/my/obspyDMT
cd /path/to/my/obspyDMT
```

obspyDMT can be installed by:

```
pip install -e .
```

or

```
python setup.py install
```

obspyDMT can be used from a system shell without explicitly calling the `Python` interpreter. The following command checks the dependencies required for running the code properly:

```
obspyDMT --check
```

obspyDMT contains various option flags for customizing the request. Each option has a reasonable default value, which the user can change to adjust obspyDMT option flags to a specific request. The following command displays all available options with their default values:

```
obspyDMT --help
```

The options are grouped by topics. To display only a list of these topic headings, use

```
obspyDMT --options
```



and to see the full help text for only one topic (e.g., group 2), use

```
obspyDMT --list_option 2
```

