# Peer review of "obspyDMT: A Python Toolbox for Retrieving and Processing of Large Seismological Datasets"

_Solid Earth, 2017_

## Referee Comment (RC1) · P. Poli (Referee) · 24 May 2017

Dear Authors and Editor,

please find my comments about the article: obspyDMT: A Python Toolbox for Retrieving and Processing of Large Seismological Datasets from Hosseini and Sigloch.

I ad the chance to work with obspyDMT for a while, but I have to admit that the new version presented here provide great advancements.

This paper illustrate in details all the tools offered by obspyDMT, and the presentation, the figures, and the table are excellent.

I have just few comments:

Table 2: for the comment –event_catalog the default is LOCAL. What it is a LOCAL catalog? It is a catalog the user has in the local directory? I guess this can be explained a little better.

I would like to see a better explanation of the outputs, in particular how the catalog is stored. The catalogs contain invaluable information for seismologists. The catalog is stored in the EVENTS-INFO directory in 3 formats, which should be explained and showed probably in the appendix. A good way to store the catalog, could be to have a simple csv file in which lat lon dep time mag mag type contributors and event id are stored. This simple file can be used by users who prefer to process the data outside python environment, and benefit from a user friendly and cross platform format.

I would like to know better how DMT communicate with ISC. In fact, ISC provide two catalogs: the reviewed one (with a ∼2years latency) and the normal bulletin containing all the events from different agencies. I suggest that while using the ISC option the user can choose in between the two available catalogs.

Still in Table 2, for the –read_catalog, you mention 3 possibilities. Not all the seismologists are familiar with the 3 formats, I suggest to add some examples in the appendix, or add references to papers or website who can help understand this formats. Again here I suggest to add another format, which is a simple CSV file with origin time of events lat lon dep and eventually mag, to facilitate the downloading data using catalog that come out from processing of different kind. This can be very helpful for people who create their own catalog for example while detecting low frequency earthquake from cross correlation or tremors. Also, will help the users to retrieve data from catalog published just in papers, which are dedicated to some particular study and do not belong to any agency.

I think that after this, the paper can be accepted without me reading it back.

[Figure]

Many thanks,

Piero Poli
* * *

---

## Author Comment (AC1) · 1 Jun 2017

**Reply to Piero Poli's comments (RC1) on "obspyDMT: A Python Toolbox for Retrieving and Processing of Large Seismological Datasets"**
* * *
*1) Table 2: for the comment –event_catalog the default is LOCAL. What it is a LOCAL catalog? It is a catalog the user has in the local directory? I guess this can be explained a little better.*

**REPLY:** Clarified as follows (Table 2):

**OLD:**
Event catalog, currently supports LOCAL, NEIC_USGS, GCMT_COMBO, IRIS, NCEDC, USGS, INGV, ISC (default: LOCAL).

**NEW:**
Event catalog, currently supports LOCAL, NEIC_USGS, GCMT_COMBO, IRIS, NCEDC, USGS, INGV, ISC (default: LOCAL).
**"--event_catalog LOCAL" searches for an existing event catalog on the user's local machine, in the EVENTS-INFO subdirectory of --datapath <PATH>. This is usually a previously retrieved catalog.**
* * *
*2) I would like to see a better explanation of the outputs, in particular how the catalog is stored. The catalogs contain invaluable information for seismologists. The catalog is stored in the EVENTS-INFO directory in 3 formats, which should be explained and showed probably in the appendix.*

**REPLY:** Clarified as follows in the caption of Figure A1:

**OLD:**
For each request, obspyDMT creates the depicted directory tree inside the user-specified directory datapath/, and arranges the retrieved data either in different event directories (for event-based requests) or in chronologically named directories (for continuous requests). It also creates a directory in which a catalog of all requested events/time spans is stored. Raw waveforms, StationXML/response files and corrected waveforms are collected in subdirectories. While retrieving the data, obspyDMT creates metadata files such as station/event location files, stored in the info/ directory of each event.

**NEW:**
For each request, obspyDMT creates the depicted directory tree inside the user-specified directory datapath/, and arranges the retrieved data either in event subdirectories (for event-based requests) or in chronologically named subdirectories (for continuous requests). It also creates a subdirectory EVENTS-INFO/ in which a catalog of all requested events or time spans is stored. **Earthquake metadata (datetime, latitude, longitude, depth, magnitude, moment tensor, source time function) is stored in CSV and QuakeML formats (files catalog.txt, catalog.ml). File catalog_table.txt**

**organizes basic event information (latitude, longitude, depth, datetime, magnitude) in a table.** Raw waveforms, StationXML/response files and corrected waveforms are collected in subdirectories. During the data retrieval process, obspyDMT also creates metadata log files about retrieved station and event files, stored in the info/ subdirectory of each event directory.
* * *
*3) A good way to store the catalog, could be to have a simple csv file in which lat lon dep time mag mag type contributors and event id are stored. This simple file can be used by users who prefer to process the data outside python environment, and benefit from a user friendly and cross platform format.*

**REPLY:** This option existed and has been emphasized in the revised text (see previous comment).
* * *
*4) I would like to know better how DMT communicate with ISC. In fact, ISC provide two catalogs: the reviewed one (with a ~2years latency) and the normal bulletin containing all the events from different agencies. I suggest that while using the ISC option the user can choose in between the two available catalogs.*

**REPLY:** An option flag to specify the ISC bulletin has been added to obspyDMT, and Table 2 has been updated accordingly:

**OLD:** -
**NEW:**
--isc_catalog <COMPREHENSIVE/REVIEWED>
Search either the COMPREHENSIVE or the REVIEWED bulletin of the International Seismological Centre (ISC).
COMPREHENSIVE: all events collected by the ISC, including most recent events that are awaiting review.
REVIEWED: includes only events that have been relocated by ISC analysts.
(default: COMPREHENSIVE). "Example: 'REVIEWED'"

Moreover, the following wording has been added to the tutorial page of obspyDMT:
* * *
**ISC catalog**

The International Seismological Centre (ISC) provides two catalogs:

- **COMPREHENSIVE bulletin** contains all events collected by the ISC, including most recent events, which are awaiting review.
- **REVIEWED bulletin** includes all events that have been reviewed and relocated by an ISC analyst.

--isc_catalog option specifies the ISC bulletin type (default: COMPREHENSIVE). Example:

obspyDMT --datapath test_isc_comprehensive --min_date 2014-01-01 --max_date 2015-01-01 --min_mag 6.0 --event_catalog ISC --isc_catalog COMPREHENSIVE --event_info --plot_seismicity

[Figure]

Retrieval result for the same request using --isc_catalog REVIEWED :

[Figure]
* * *
*5) Still in Table 2, for the –read_catalog, you mention 3 possibilities. Not all the seismologists are familiar with the 3 formats, I suggest to add some examples in the appendix, or add references to papers or website who can help understand this formats.*

**REPLY:** Agreed, we have added wording to Table 2, as explained in response to item 6) below.
* * *
*6) Again here I suggest to add another format, which is a simple CSV file with origin time of events lat lon dep and eventually mag, to facilitate the downloading data using catalog that come out from processing of different kind. This can be very helpful for people who create their own catalog for example while detecting low frequency earthquake from cross correlation or tremors. Also, will help the users to retrieve data from catalog published just in papers, which are dedicated to some particular study and do not belong to any agency.*

**REPLY:** This is a good idea. obspyDMT version v2.0.2 can now read a CSV file as an event catalog, as detailed in the tutorial. We have added wording to Table 2 addressing comments 5 & 6:

**OLD** (Table 2)**:**
Read in an existing local event catalog and proceed.
Currently supported catalog metadata formats:
"QUAKEML", "NDK", "ZMAP".

**NEW** (Table 2)**:**
Read in an existing local event catalog and proceed. Currently supported catalog metadata formats:
"**CSV",** "QUAKEML", "NDK", "ZMAP".
**Format of the plain text CSV (comma-separated values) is explained in the obspyDMT tutorial. Refer to obspy documentation for details on QuakeML, NDK and ZMAP formats.**

Also the following wording has been added to the tutorial page of obspyDMT:
* * *
**Read an existing local event catalog**

--read_catalog <PATH> option flag reads in an existing event catalog located at <PATH> and proceeds. Currently supported catalog metadata formats: "CSV", "QUAKEML", "NDK", "ZMAP"
(Refer to obspy documentation for details on QuakeML, NDK and ZMAP formats).

**CSV format:** obspyDMT can read a CSV file as an event catalog. This must be a list of comma-separated values containing some or all of the fields below, one event per line:

event_number,event_id,datetime,latitude,longitude,depth,magnitude,magnitude_type,author,flynn_region,mrr,mtt,mpp,mrt,mrp,mtp,stf_func,stf_duration

File "catalog.txt", generated by obspyDMT in EVENTS-INFO subdirectory provides an example of such a file.

Example:

**number,event_id,datetime,latitude,longitude,depth,magnitude,magnitude_type,author,flynn_region,**
mrr,mtt,mpp,mrt,mrp,mtp,stf_func,stf_duration
1,20110311_054623.a,2011-03-11T05:46:23.200000Z,38.2963,142.498,19.7,9.1,MW,None,NAN,No
ne,None,None,None,None,None,triangle,164.914

datetime, latitude, longitude, depth and magnitude are mandatory. Optional fields may be set to "None", as in the following example where only datetime, latitude, longitude, depth and magnitude are set:

**number,event_id,datetime,latitude,longitude,depth,magnitude,magnitude_type,author,flynn_region,**
mrr,mtt,mpp,mrt,mrp,mtp,stf_func,stf_duration
1,None,2011-03-11T05:46:23.200000Z,38.2963,142.498,19.7,9.1,None,None,None,None,None,None
,None,None,None,None,None

---

## Short Comment (SC1) · 9 Jun 2017

Dear Authors,

Thank you for this article, it gives a very handy insight into the obspyDMT use.

I think in your appendix B (p32) line 16, you may have made a mistake by writing "T(jw) in eq B3" instead of "G(jw) in eq B3". Because you called G(jw) your frequency response function in the upper part of this appendix.

Thank you again for all those information about obspyDMT.

Kind regards,

[Figure]

Frédéric Dubois

---

## Author Comment (AC2) · 12 Jun 2017

Dear Frédéric Dubois,

Thank you for your careful reading of our manuscript and for pointing out this inconsistency. We have replaced "T(jw) in eq B3" by "G(jw) in eq B3" (Appendix B, page 32, line 16).

Kind regards,

[Figure]

Kasra Hosseini & Karin Sigloch

---

## Referee Comment (RC2) · R. Zaccarelli (Referee) · 29 Jun 2017

Dear Authors and Editor,

please find my comments about the article: obspyDMT: A Python Toolbox for Retrieving and Processing of Large Seismological Datasets from Hosseini and Sigloch.

As I am responsible for the development of scientific software and web infrastructures, and thus I often work with obspy, I found the article extremely interesting and well explained.

I do not have major comments, just one minor one: On section 3.8 (line - or paragraph

[Figure]

- 25) the sentence "Although this file is fully customizable" sounds a bit misleading or needs clarification:

- If the authors mean that the processing steps can be achieved by flags and - for more experts users - the python file in the package is always editable, I would remove or rephrase the sentence for two reasons: first, modifying open source code (let alone licensing for the moment) is a feature common to any open source software and I do not see why it is worth a particular mention. Second, it might be interpreted like the authors encouraged the modification of their own source code, which sounds quite odd (and exposes any user to potentially breaking the code, at their own risk).

- On the other hand, if a particular customization has been implemented, maybe the sentence might deserve a little bit more explanation

Let alone this minor detail, I think that the paper can be accepted without any further modification

Best regards

Riccardo Zaccarelli

---

## Author Comment (AC3) · 25 Jul 2017

**Reply to Riccardo Zaccarelli's comments (RC2) on "obspyDMT: A Python Toolbox for Retrieving and Processing of Large Seismological Datasets"**

*1) On section 3.8 (line - or paragraph - 25) the sentence "Although this file is fully customizable" sounds a bit misleading or needs clarification:*
*- If the authors mean that the processing steps can be achieved by flags and - for more experts users - the python file in the package is always editable, I would remove or rephrase the sentence for two reasons: first, modifying open source code (let alone licensing for the moment) is a feature common to any open source software and I do not see why it is worth a particular mention. Second, it might be interpreted like the authors encouraged the modification of their own source code, which sounds quite odd (and exposes any user to potentially breaking the code, at their own risk).*
*- On the other hand, if a particular customization has been implemented, maybe the sentence might deserve a little bit more explanation*

**REPLY:** Thank you, clearly this sentence was too short and cryptic. We are not talking about changing source code, as hopefully our new wording makes clear. Rather we are enabling users to customize their waveform processing via scripting calls to external programs such as obspy and SAC. Virtually any processing tool which can be called from the command line (e.g., obspy, SAC or a combination of these tools) can be integrated, by writing a processing script **my_proc_unit** and executing is via the obspyDMT call **--pre_process my_proc_unit**. Currently two example scripts are also provided, located at /path/to/my/obspyDMT/obspyDMT.

**OLD:**
Although this file is fully customizable, several common processing steps can be done via options flags (without changing/writing new processing instructions).

**NEW:**
This scripting file can be freely edited by the user and may include calls to external waveform processing programs such as obspy or SAC. This vastly expanding the possibilities for waveform processing and lets users easily adapt and integrate functionality from earlier, non-obspyDMT work flows. Instructions in this file are written at the waveform level, and obspyDMT applies them to all waveforms in the entire data set (in serial or in parallel mode). The default file included in the current distribution, /path/to/my/obspyDMT/obspyDMT/process_unit.py, can perform routine processing steps such as resampling, data format conversion, and instrument correction. These steps can be accessed via dedicated option flags, each of which results in the execution of only the appropriate part of processing script process_unit.py (see --pre_process option flag). Hence a user requiring only these routine operations need not create or modify a processing script file.